# Downregulated Expression of miR-200c-3p in Plasma Exosome as a Potential Biomarker in Takayasu’s Arteritis

**DOI:** 10.3390/ijms26072881

**Published:** 2025-03-22

**Authors:** Lihong Du, Chenglong Fang, Biqing Huang, Xiaofeng Zeng, Jing Li, Xinping Tian

**Affiliations:** 1Department of Rheumatology and Clinical Immunology, Peking Union Medical College Hospital, Chinese Academy of Medical Sciences, Peking Union Medical College, Beijing 100006, China; lihondu@163.com (L.D.); fangcl900523@126.com (C.F.); byhuangbiqing@student.pumc.edu.cn (B.H.); zengxfpumc@163.com (X.Z.); 2National Clinical Research Center for Dermatologic and Immunologic Diseases (NCRC-DID), Ministry of Science & Technology, Beijing 100006, China; 3State Key Laboratory of Complex Severe and Rare Diseases, Peking Union Medical College Hospital, Beijing 100006, China; 4Key Laboratory of Rheumatology and Clinical Immunology, Ministry of Education, Beijing 100730, China

**Keywords:** biomarker, exosomes, MiRNAs, miR-200c-3p, Takayasu’s arteritis

## Abstract

Our previous work identified several differentially expressed miRNAs (DEmiRNAs) in plasma exosomes from Takayasu’s arteritis (TAK) patients. This study aimed to validate these findings and explore the correlation between DEmiRNAs and clinical parameters in untreated TAK. Plasma exosomes were isolated from 30 untreated TAK patients and 20 healthy controls. qPCR was used to quantify miR-34a-5p, miR-143-3p, miR-22-3p, miR-200c-3p, and miR-21-5p expression. Correlations between miRNA levels, clinical data, inflammation markers, and T helper cell frequencies were analyzed. The target genes of validated DEmiRNAs were identified using mirDIP, and pathway enrichment analysis was performed using GO/KEGG. The effect of validated DEmiRNAs on the MAPK pathway and proliferation in human aortic endothelial cells (HAECs) was investigated in vitro. Only miR-200c-3p expression was validated as significantly downregulated in plasma exosomes from untreated TAK patients. Lower miR-200c-3p levels correlated negatively with ITAS-2010 scores and were associated with relapsed disease. MiR-200c-3p levels also negatively correlated with circulating Th17.1 cell frequencies. In vitro, the TAK exosome treatment activated ERK1/2 and JNK pathways and promoted HAEC proliferation, which was inhibited by the miR-200c-3p mimic. The pathway enrichment analysis showed that the MAPK pathway may be involved. This study confirms the reduced miR-200c-3p expression in plasma exosomes from TAK patients, suggesting its potential as a biomarker for vascular inflammation. MiR-200c-3p may exert protective effects in TAK by suppressing MAPK pathway activation and EC proliferation.

## 1. Introduction

Takayasu’s arteritis (TAK) is a female-predominant systemic large-vessel vasculitis (LVV) that usually has an early onset at a young age. TAK is characterized by granulomatous inflammation affecting the aorta and its primary branches, resulting in segmental stenosis, occlusion, dilatation, or aneurysm, causing substantial morbidity and mortality [1].

The complex interactions between immune cells and vascular cells play critical roles in the development of TAK [2]. These interactions contribute to the inflammatory cascade and vascular remodeling that define the disease. MicroRNAs (miRNAs), small non-coding RNA molecules that regulate gene expression, have emerged as key players in a variety of inflammatory and vascular diseases [3]. Specifically, miRNAs are known to influence endothelial cell function, immune cell activation, and vascular smooth muscle cell proliferation, and these processes are all implicated in TAK [3]. For example, miRNAs have been shown to modulate inflammatory cytokine production in other forms of vasculitis, such as Behçet’s disease [4]. Furthermore, the dysregulation of miRNAs has been implicated in the pathogenesis of atherosclerosis, where they influence vascular smooth muscle cell proliferation and plaque stability [5]. These findings suggest a potential role for miRNAs in the pathogenesis of TAK.

Exosomes, as key mediators of intercellular communication, are present in various body fluids and contain miRNAs and proteins that can be transferred between cells. Exosome-derived miRNAs have shown potential as disease biomarkers in a variety of pathological conditions, including lupus nephritis [6]. However, research specifically focused on the role of exosomal miRNAs in TAK is limited.

In our previous study, five differentially expressed miRNAs (DEmiRNAs) in plasma exosomes isolated from 10 untreated TAK patients had been identified, including miR-34a-5p, miR-143-3p, miR-22-3p, miR-200c-3p, and miR-21-5p [7]. Furthermore, our recent publication [8] demonstrated that exosome-derived miR-199a-5p plays a role in vascular remodeling and inflammatory infiltration in TAK, providing further evidence for the involvement of miRNAs in this disease. Herein, we verified the expression profile of these DEmiRNAs in the validation cohort consisting of 30 treatment-naïve TAK patients and established the correlation between DEmiRNAs and clinical parameters. The targeted genes and potential functions of DEmiRNAs were investigated.

## 2. Results

### 2.1. Downregulated Expression of miR-200c-3p in Plasma Exosome Was Validated in Untreated TAK Patients

Isolated plasma exosomes from TAK patients and healthy individuals were verified by characteristic morphology in TEM (shown in Figure 1A) and protein expression profile including the presence of CD9, CD63, and TSG101, and the lack of calnexin expression (shown in Figure 1B). The presence of a faint ApoA1 band was also checked (shown in Appendix A) to assess the level of lipoprotein contamination in the exosome preparations. Compared with 20 age- and sex-matched healthy individuals, miR-200c-3p expression was downregulated in 30 treatment-naïve TAK patients. In contrast, there was no significant difference found in the expression levels of miR-34a-5p, miR-143-3p, miR-22-3p, and miR-21-5p between patients and healthy controls (shown in Figure 1C). Therefore, the qPCR values for miR-200c-3p in TAK patients and healthy controls were used to generate a ROC curve to further evaluate its diagnostic potential (shown in Appendix A). The corresponding area under the curve (AUC) was 0.908, indicating high discriminatory ability.

### 2.2. Lower Levels of miR-200c-3p in Plasma-Derived Exosomes Were Associated with Vascular Inflammation and Th1 Response

In the correlation analysis, the expression levels of miR-200c-3p in plasma exosomes from untreated TAK patients (patients newly diagnosed with TAK who have not yet received any glucocorticoids, immunosuppressants, or surgical treatment for their condition) were negatively correlated with the ITAS-2010 score at the first visit (shown in Figure 2A). Of note, no correlation between systemic inflammation biomarkers (including serumal CRP, TNF-α, IL-6, and ESR) and exosomal miR-200c-3p expression levels was observed in these patients (shown in Figure 2B–E). The Kerr score was used to assess disease activity, and the patients with disease relapse during follow-up exhibited relatively lower expression levels of miR-200c-3p in plasma exosomes at baseline (shown in Figure 2F). The Kaplan–Meier analysis also suggested that the patients with a lower expression level of miR-200c-3p at baseline were prone to relapse (shown in Figure 2G). In addition, we found that only the circulating frequencies of Th17.1 cells, rather than those of Th1, Th17, and Th2 cells, were negatively correlated with miR-200c-3p expression levels in plasma exosomes (shown in Figure 3A–D).

### 2.3. Enrichment Analysis of Targeted Genes of miR-200c-3p and In Vitro Experiments Suggested the Possible Pathogenic Role of the MAPK Pathway in TAK

To further investigate the role of miR-200c-3p in TAK pathogenesis, we analyzed and identified a total of 950 targeted genes of miR-200c-3p with “high” to “very high” confidence by the mirDIP database. The GO enrichment analysis suggested these candidate genes were associated with cell junction assembly, cell–cell junction, and transcription coregulator activity (shown in Figure 4A,B).

The KEGG enrichment analysis suggested the targeted genes were related to the MAPK pathway (shown in Figure 5A,B). In vitro cell culture experiments were conducted to explore the effects of miR-200c-3p in TAK plasma exosome-treated HAECs. These cells exhibited activated the ERK1/2 pathway and JNK pathway, but not the p38 MAPK pathway (shown in Figure 5C), compared with those cultured with plasma exosomes from healthy donors. This effect could be suppressed by miR-200c-3p mimic supplementation. The quantitative densitometry analysis of the Western blot images is presented in Appendix A. In the EdU assay, the treatment with plasma exosomes from healthy donors slightly enhanced HAEC proliferation, and this effect was further augmented by the addition of a miR-200c-3p inhibitor (Figure 5D). The treatment with plasma exosomes from TAK patients led to the enhanced proliferation of HAECs in a dose-dependent manner, while the addition of the miR-200c-3p mimic or the JNK/ERK1/2 pathway inhibitor, astragaloside IV, downregulated TAK-plasma-exosome-induced HAECs proliferation (shown in Figure 5E). In addition, the JNK/ERK1/2 pathway activator dehydrocrenatine attenuated the effects of the miR-200c-3p mimic. Altogether, miR-200c-3p in plasma exosomes might exert protective effects via suppressing MAPK pathway activation in ECs and EC proliferation (shown in Figure 5E).

## 3. Discussion

In the current study, we analyzed an independent validation cohort consisting of 30 untreated TAK patients to verify differentially expressed microRNA (DEmiRNAs) in plasma exosomes from TAK patients. Previously, a panel of five exosomal microRNAs had been identified using paired-end sequencing. We found that miR-200c-3p was the only DEmiRNAs in plasma exosomes between TAK patients and the healthy control based on the results from our previous miRNA genomics analysis [7]. To the best of our knowledge, this is the first attempt to investigate the association between vascular inflammations of TAK and plasma exosome miRNA.

The discrepancy between systemic inflammation biomarkers and vascular inflammation is commonly seen in TAK. A reliable blood biomarker for detecting vascular inflammation is still urgently needed for low invasiveness and a relatively low cost. Pentraxin-3 was suggested as a more promising biomarker to reflect vascular disease activity in TAK. However, in another study, PTX3 failed to indicate smoldering vascular wall inflammation [9]. In this study, we found that the expression level of miR-200c-3p in plasma exosomes was negatively correlated with disease activity, which was reflected by the ITAS-2010 score, and the ITAS-2010 mainly assesses the clinical manifestation of vascular inflammation, stenosis, and occlusion [10]. Elevated ITAS-2010 scores may be an independent risk factor for vascular wall inflammation in TAK patients [11]. This association, coupled with the finding that lower baseline miR-200c-3p levels were predictive of disease relapse, suggests a potential link between miR-200c-3p and disease activity, as measured by ITAS-2010. Interestingly, no correlation between miR-200c-3p expression levels and systemic serum inflammation biomarker levels was observed in untreated TAK. Taken together, these results suggested that miR-200c-3p may be specifically associated with vascular inflammation in TAK.

Several pieces of evidence had suggested the role of miR-200c-3p in vascular disease and inflammation. Mao Y et al. reported that MiR-200c-3p promotes ox-LDL-induced an endothelial-to-mesenchymal transition in human umbilical vein endothelial cells [12]. In the context of artery bypass grafts, miR-200c-3p promotes an endothelial-to-mesenchymal transition and neointimal hyperplasia [13]. A significant negative correlation between the expression of miR-200c-3p and inflammation and vascular health was found in type 1 diabetes mellitus [14]. In COVID-19 and cholestatic liver fibrosis, miR-200c-3p has the potential to negatively regulate IL-6 expression, which is a main component of the cytokine storm [15,16]. Together, these findings suggest that miR-200c-3p exerts diverse biological effects that depend on the cell type and context. In this study, we found that miR-200c-3p expression levels in plasma exosomes were negatively correlated with circulating Th17.1 cell frequencies; this finding suggests a potential relationship between miR-200c-3p and the Th1 response, particularly given that Th17.1, while elevated in active TAK, is known to contribute to granuloma formation through IFN-γ secretion and corticosteroid resistance via p-glycoprotein expression [17]. Consistent with prior research demonstrating the role of th17.1 in regulating Th1 responses, our results further highlight the potential of exosomal miR-200c-3p as a key modulator of Th1-mediated immunity in TAK, providing a novel avenue for understanding disease pathogenesis. Of note, the Th1 response, rather than the Th17 response, is associated with vascular wall inflammation to a greater extent [18]. In addition, previously published data suggest that the Th1 response may dominate in shaping vascular wall inflammation in TAK [18]. Whether manipulating miR-200c-3p expressions is plausible to modulate the inflammatory process in TAK needs to be further investigated.

The enrichment analysis of targeted genes of miR-200c-3p suggested the association with the MAPK pathway. There is no direct evidence for the pathogenic role of the MAPK pathway in the development of TAK. However, the association between the MAPK pathway (especially ERK1/2) and endothelial cell proliferation has been well-established in the context of vascular diseases [19,20]. Persistent vascular inflammation in TAK would drive the proliferation of endothelial cells, leading to intimal hyperplasia and vascular occlusion [21]. In the present study, our results provided the rationale that miR-200c-3p might inhibit the TAK-plasma-exosome-treatment-induced HAECs proliferation via suppressing the activation of ERK1/2 and JNK pathways (Figure 5D,E). Collectively, these findings suggest that the reduction in the exosomal miR-200c-3p level might be involved in the vascular remodeling process of TAK.

However, this study is subject to certain limitations. Firstly, the relatively small overall sample size restricts the statistical power of our analysis and the generalizability of our findings, due to the low incidence rate of TAK (40/million per year) [2]. The second limitation concerns the highly skewed sex distribution within our cohort, specifically the inclusion of only three men. While the miR-200c-3p levels in these individuals did not appear as outliers compared to the other 27 women, the substantial imbalance prevents us from drawing definitive conclusions about sex-specific differences. Furthermore, although the observed sex ratio (of man to woman) of 3:27 aligns with the predominantly female demographic of TAK [2,22], suggesting a degree of representativeness, this does not eliminate the potential for gender bias. Nevertheless, the observed consistency in miR-200c-3p levels, despite the limited sample from men, hints at the possibility of miR-200c-3p serving as a valuable biomarker in TAK regardless of sex. This warrants further investigation in larger, sex-stratified studies.

## 4. Materials and Methods

### 4.1. Patients and Healthy Controls

Patients with untreated TAK (n = 30) and age- and sex-matched healthy donors (n = 20) were included in this study. Patients with TAK fulfilled the ACR 1990 classification criteria for TAK [23]. Patient characteristics, including demographic data, clinical features, and disease duration, are summarized in Table 1. The median disease duration was 15.5 months. Elevated levels of CRP or ESR were observed in 26 patients, as shown in Table 1. Disease activity state was assessed by the Kerr score [24], Indian Takayasu Clinical Activity Score (ITAS-2010) [11], and EULAR consensus definitions in LVV [25]. TAK-associated biomarkers of inflammation including C-reactive protein (CRP), erythrocyte sedimentation rate (ESR), tumor necrosis factor-α (TNF-α), and interleukin-6 (IL-6) were collected. Vascular progression was determined by clinical manifestation and imaging findings. It was defined as the first recurrence of active disease within one year as a disease relapse. All TAK patients were followed in the National Clinical Research Center for Dermatologic and Immunologic Diseases (NCRC-DID), Ministry of Science & Technology, State Key Laboratory of Complex Severe and Rare Diseases in Peking Union Medical College Hospital. This study was performed according to the principles of the Declaration of Helsinki and informed consent was obtained from all subjects, and the protocol was reviewed and approved by the Medical Ethics Committee of Peking Union Medical College Hospital, approval number S-478. Written informed consent was received from all participants in this study.

### 4.2. Plasma Preparation and Total Exosome Isolation

Whole blood was collected from TAK patients and healthy individuals. Plasma was isolated and stored at −80 °C until analysis. For exosome isolation, plasma samples were thawed at 37 °C and filtered using 0.8 μm membranes. Plasma exosomes were concentrated using a total exosome isolation kit (Thermo Fisher Scientific, Waltham, MA, USA) according to the technical manual. Isolated plasma exosomes were verified using a transmission electron microscope (TEM) and the immunoblotting of calnexin, CD63, CD9, and TSG101.

### 4.3. MicroRNA Validation by Quantitative Polymerase Chain Reaction (qPCR)

MicroRNA in plasma exosomes was extracted using miRNeasy Mini Kit (Qiagen GmbH, Hilden, Germany). Exogenous cel-mir-39-3p was used as the standard RNA for calibration. The expression levels of miR-34a-5p, miR-143-3p, miR-22-3p, miR-200c-3p, and miR-21-5p were determined by qPCR using a Mir-X miRNA First-Strand Synthesis Kit and Mir-X miRNA qRT-PCR TB Green Kit (Takara Biotechnology (Dalian) Co., Ltd., Dalian, China), following the manufacturer’s instructions. qPCR reactions were performed with Bio-rad CFX96 (Bio-Rad Laboratories, Inc., Hercules, CA, USA). Gene expression was calculated using the 2^−ΔΔCt^ method. All primers were provided by RIBOBIO (Guangzhou, China).

### 4.4. Peripheral Blood Mononuclear Cells (PBMCs) Collection and Analysis by Flow Cytometry

PBMCs were freshly isolated from TAK patients and healthy controls by density gradient centrifugation for immunophenotyping using flow cytometry. For T helper (Th) cells subgroup analysis, including Th1 cells (defined as CD4+CXCR3+CCR6−), Th2 cells (defined as CD4+CXCR3−CCR6−), Th17 cell (defined as CD4+CXCR3−CCR6+), and Th17.1 cells (defined as CD4+CXCR3+CCR6+), samples were stained with APC/Cyanine7 anti-human CD4, FITC anti-human CCR6, and PerCP/Cyanine5.5 anti-human CXCR3 for 30 min before analyzed by flow cytometry.

### 4.5. Cell Culture

The human aortic endothelial cells (HAECs) used in the study were purchased from American Type Culture Collection (ATCC; Manassas, VA, USA), and were cultured in an endothelial cell medium (Sciencell; San Diego, CA, USA) supplemented with 5% exosome-depleted fetal bovine serum (Absin Bioscience Inc., Shanghai, China). HAECs were further treated with plasma exosomes from TAK patients or healthy individuals. The inhibitor and mimic of validated DEmiRNAs were synthesized by RIBOBIO (Guangzhou, China) and transfected to the cultured HAECs using riboFECT™ CP Transfection Reagent (Guangzhou, China) to investigate the effects of DEmiRNAs mimic and inhibitor.

### 4.6. Western Blot

Protein extracts from HAECs were subjected to electrophoresis (SDS-PAGE) before being transferred onto a polyvinylidene difluoride membrane and incubated with an anti-Phospho-p44/42 MAPK (Erk1/2) (Thr202/Tyr204) antibody (Cell Signaling Technology; Boston, MA, USA), anti-Phospho-SAPK/JNK (Thr183/Tyr185) antibody (Cell Signaling Technology; Boston, MA, USA), anti-Phospho-p38 MAPK (Thr180/Tyr182) antibody (Cell Signaling Technology; Boston, MA, USA), and anti-GADPH antibodies (Proteintech Group, Inc., Wuhan, China). Immunoreactive bands were visualized using appropriate HRP-conjugated secondary antibodies (Cell Signaling Technology; Boston, MA, USA). Images were acquired using the Tanon 5800 system (Tanon Science & Technology Co., Ltd., Shanghai, China) and analyzed using ImageJ version 1.53T (National Institutes of Health, Bethesda, MD, USA).

### 4.7. EdU Assay

The EdU assay of HAECs was performed using a BeyoClick™ EDU Cell Proliferation Kit with TMB (Beyotime Biotechnology, Shanghai, China) according to the manufacturer’s instructions. In brief, HAECs were seeded in 96-well plates at a density of 1000 cells per well and received different treatments. The exosome concentrations of 100, 1000, and 10,000 EVs/cell were determined using Nanoparticle Tracking Analysis (NTA) on a Malvern NanoSight NS300 (Malvern Panalytical; Malvern, UK). The isolated exosomes were diluted and analyzed according to the manufacturer’s instructions. The resulting particle counts were then normalized to the number of cells used for exosome isolation to determine the EVs/cell ratio. The exosome treatment was administered by directly adding the isolated exosomes to the cell culture medium. The exosomes were resuspended in the HAEC medium and added to the cells at the indicated concentrations (100, 1000, or 10,000 EVs/cell). The medium was not changed during the treatment period. HAECs were subsequently incubated with 10 μM EdU for 2 h at 37 °C, followed by fixation with 4% paraformaldehyde and permeabilization with 0.3% Triton X-100. After incubation in a 50 μL Click reaction buffer for 30 min at room temperature in the dark, a Streptavidin-HRP working solution was added and incubated for another 30min at room temperature. After 5 min of development with the TMB developing buffer, the absorbance was measured at 640 nm using a Synergy H1 microplate reader (BioTek; Winooski, VT, USA).

### 4.8. Statistical Methods

Quantitative variables were distributed non-normally and presented with the median and range, and the differences between groups were compared by the Mann–Whitney test. To evaluate the diagnostic potential of miR-200c-3p in distinguishing Takayasu’s arteritis (TAK) patients from healthy controls (HC), Receiver Operating Characteristic (ROC) curve analysis was performed using the qPCR-derived miR-200c-3p expression values. The ROC curve was generated by plotting the sensitivity against 1-specificity at various threshold levels. The area under the ROC curve (AUC) was calculated to assess the overall discriminatory ability of miR-200c-3p. The optimal diagnostic threshold for miR-200c-3p was determined based on the point on the ROC curve that maximized the Youden’s J statistic (Youden’s index = sensitivity + specificity − 1). This threshold was then used to calculate the sensitivity and specificity of miR-200c-3p for differentiating TAK patients from healthy controls. The correlation between gene expression levels was presented by the Pearson correlation coefficient. The gene expression levels were measured and arranged in a descending order; higher values correspond to a higher expression, and the top 6 (50%) were considered as having a “high” expression level, while the last 6 (50%) were considered as having a “low” expression level. Kaplan–Meier survival curves were performed to compare the relapse probability between the high and low miR-200c-3p gene expression groups for one year, and the difference in the time between the two groups was tested using a log-rank test. All statistical tests were two-tailed with a significance threshold of 0.05. Statistical analysis was performed using GraphPad Prism version 8.0.1 for Windows (GraphPad Software, San Diego, CA, USA). The target genes of DEmiRNAs were predicted by the microRNA Data Integration Portal (mirDIP, Version 5.2.8.1, Database version 5.2.3.1; November 2022) (http://ophid.utoronto.ca/mirDIP (accessed on 28 December 2022)) and the top 5% targeted genes with “high” to “very high” confidence were chosen to conduct the enrichment analysis, which was performed by using Gene Ontology (GO) and Kyoto Encyclopedia of Genes and Genomes (KEGG) databases. The enrichment results were visualized using the R (V4.0.4) Package clusterProfiler V.4.4.4, ggplot2 V3.4.0, GOplot V1.0.2, circlize V0.4.15, and ComplexHeatmap V2.12.1.

## 5. Conclusions

Our findings confirmed the reduced expression of miR-200c-3p in plasma exosomes from TAK patients and its potential as a biomarker of vascular inflammation in TAK, albeit it remains undetermined whether miR-200c-3p contributes to the pathogenesis via MAPK pathway activation.

## Figures and Tables

**Figure 1 ijms-26-02881-f001:**
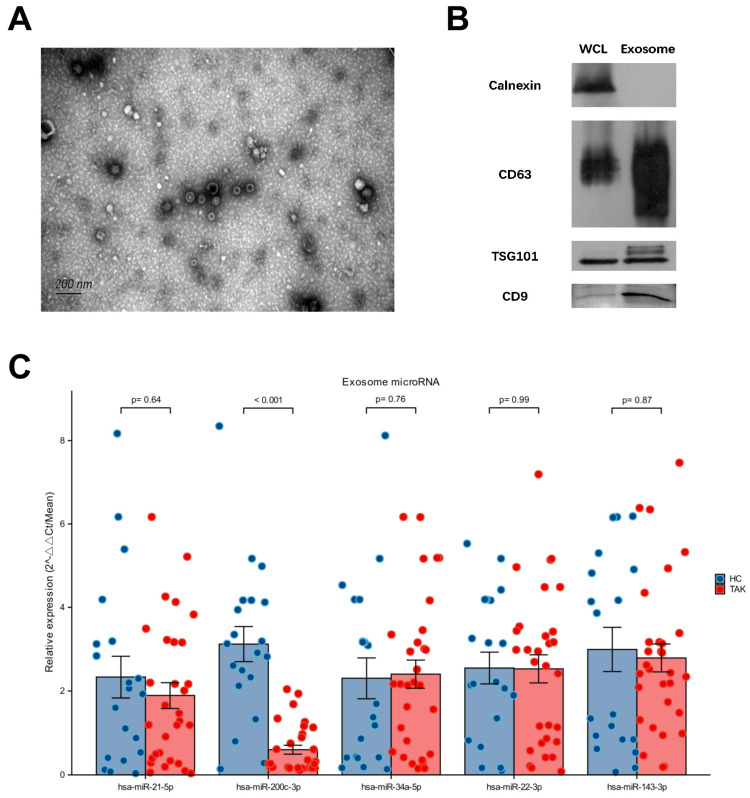
Identification of plasma exosome and candidate miRNA level was detected by qRT-PCR. (**A**). TEM micrographs of plasma exosome (scale bar = 200 nm). (**B**). Western blot analysis for positive exosome markers (CD63, TSG101, and CD9) and negative exosome marker (Calnexin). (**C**). qRT-PCR analysis of miR-21-5p, miR-200c-3p, miR-34a-5p, miR-22-3p, and miR-143-3p from plasma exosomes of 30 TAK patients and 20 healthy controls. Of these five miRNAs, only miR-200c-3p showed a statistically significant difference between the two groups. TEM, transmission electron microscopy; WCL, whole cell lysate; miRNA, microRNA; qRT-PCR, quantitative RT-PCR; TSG101, tumor susceptibility gene 101 protein.

**Figure 2 ijms-26-02881-f002:**
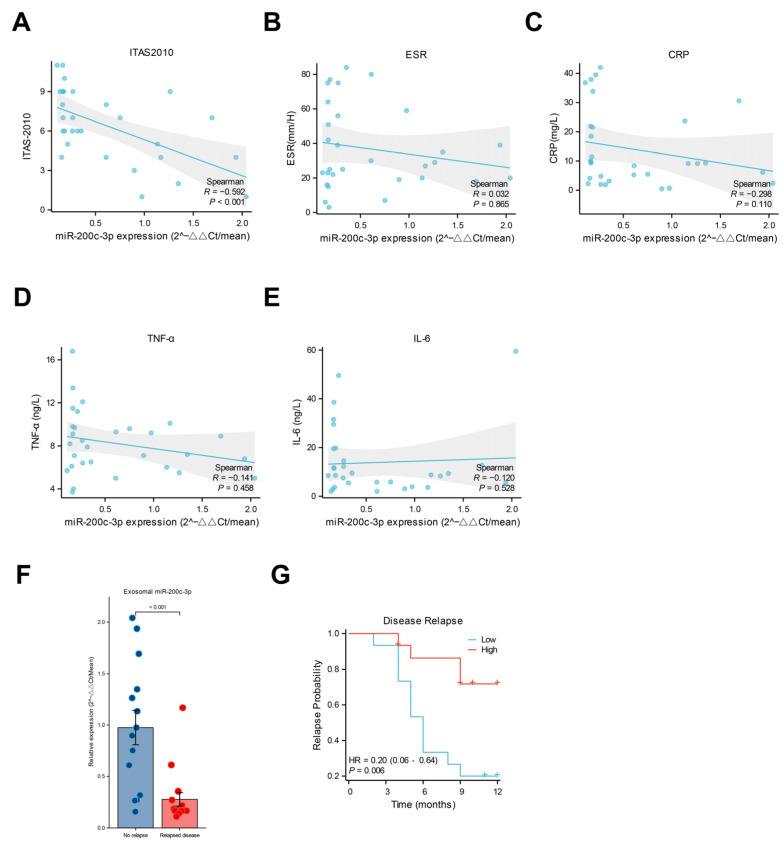
Clinical relevance analysis. (**A**–**E**). Correlation analysis of miR-200c-3p and ITAS-2010 score, CRP, ESR, TNF-α, and IL-6 levels respectively. The grey shade represents the 95% confidence interval. Correlations were assessed using Spearman’s correlative analysis. (**F**). Levels of miR-200c-3p between relapse and non-relapsed groups. (**G**). Kaplan–Meier estimates of time to relapse probability of TAK patients according to levels of miR-200c-3p. ITAS-2010, Indian Takayasu Arteritis Activity Score; CRP, C-reactive protein; ESR, erythrocyte sedimentation rate; TNF-α, tumor necrosis Factor-α; IL-6, interleukin 6.

**Figure 3 ijms-26-02881-f003:**
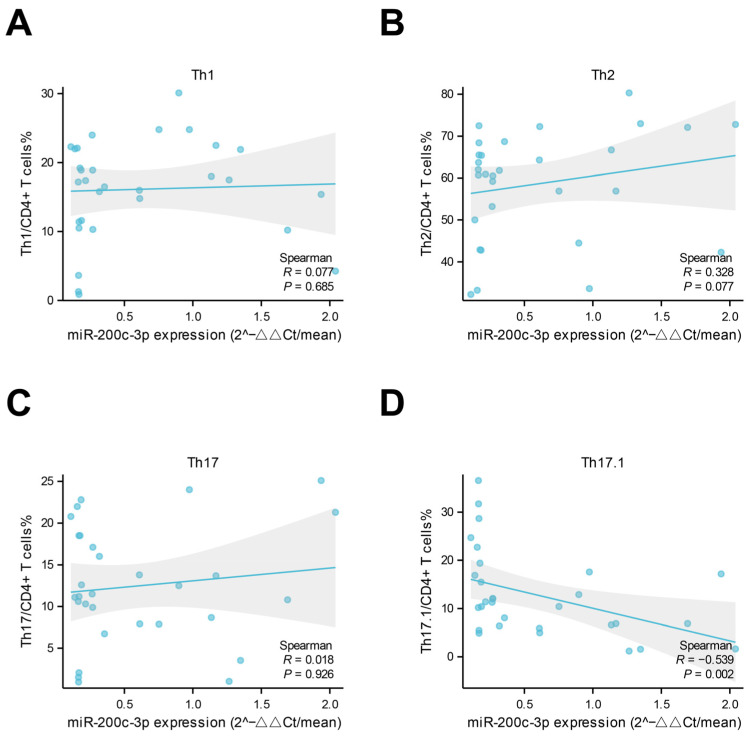
Spearman correlation analysis of miR-200c-3p expression in plasma exosomes and circulating frequencies of T helper cell subsets in Takayasu’s arteritis patients. (**A**–**D**). Correlation analysis was performed between miR-200c-3p levels and the levels of Th1, Th2, Th17, and Th17.1 cells, respectively. The grey shade represents the 95% confidence interval. No significant correlations were observed between miR-200c-3p and Th1, Th2, or Th17 cells. However, a moderate negative correlation was found between miR-200c-3p and Th17.1 cell levels (R = −0.539). Circulating frequencies of T helper cell subgroups of TAK patients were determined via flow cytometry.

**Figure 4 ijms-26-02881-f004:**
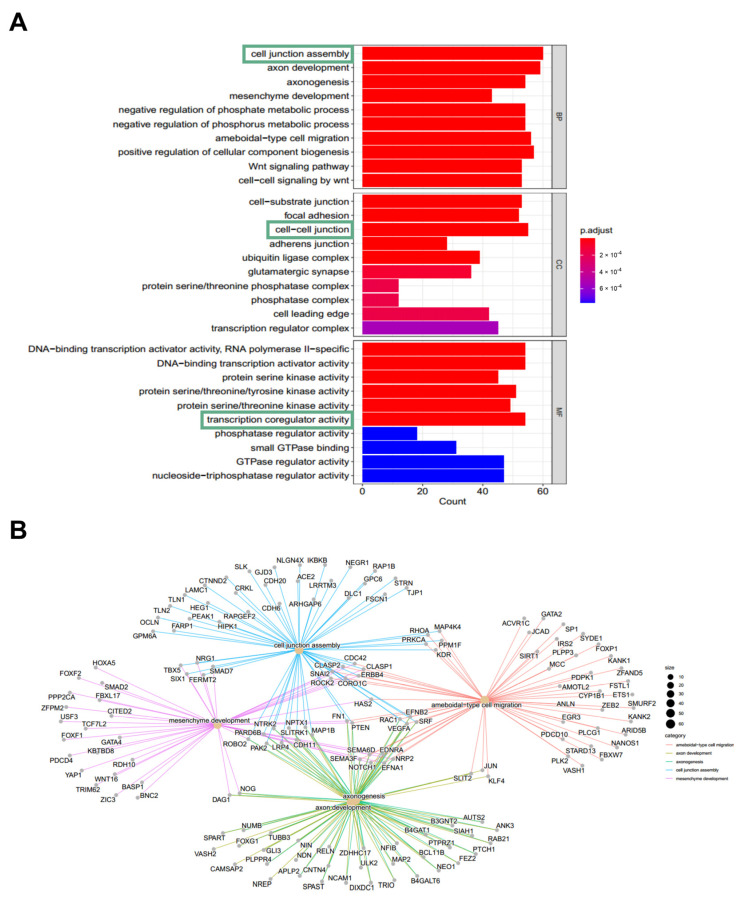
GO enrichment analysis. (**A**). The top 10 enriched items in GO-BP, GO-CC, and GO-MF for target genes of miR-200c-3p, respectively. Green box highlights the most significant GO terms. (**B**). GO-BP functional networks of upregulated targeted genes; color of the node represented significant GO terms. GO, Gene Ontology; GO-BP, GO biological process; GO-CC, GO cellular component; GO-MF, GO molecular function.

**Figure 5 ijms-26-02881-f005:**
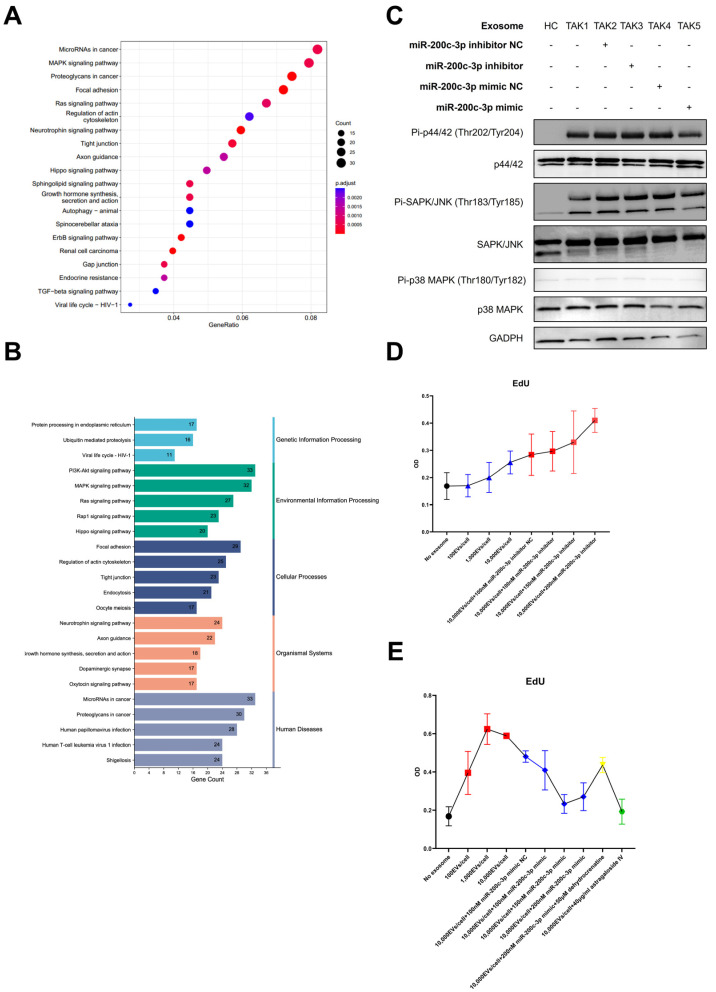
KEGG enrichment analysis, differential expression validation, and functional assay. (**A**). The top 20 enriched KEGG terms are shown. Each dot represents one KEGG gene set. (**B**). The top 5 significant secondary classifications of KEGG include genetic information processing, environmental information processing, cellular process, organismal systems, and human diseases. KEGG, Kyoto Encyclopedia of Genes and Genomes. (**C**). Phosphorylated p44/42 (ERK 1/2 pathway), phosphorylated JNK (JNK pathway), phosphorylated p38 MAPK (p38 pathway), and GAPDH immunoblotting of HAECs lysates. Treatment with TAK exosomes or a miR-200c-3p inhibitor increased phosphorylation of p44/42 and JNK, while a miR-200c-3p mimic reduced their phosphorylation. Phosphorylation of p38 MAPK was not significantly affected by any treatment. TAK-labeled lanes represent plasma exosomes isolated from different Takayasu’s arteritis (TAK) patients. The grouping of gels/blots cropped from different parts of the same gel. (**D**). EdU incorporation assay assessing the proliferation of HAECs. Cells were treated with plasma exosomes from healthy control donors (HC) or a miR-200c-3p inhibitor. Both treatments resulted in a significant increase in HAEC proliferation. Different colors indicate the different treatment types. (**E**). EdU incorporation assay assessing HAEC proliferation. Cells were treated with plasma exosomes from TAK patients (TAK), a miR-200c-3p mimic, dehydrocrenatine (50 μM; JNK/ERK1/2 activator), or astragaloside IV (40 μg/mL; JNK/ERK1/2 inhibitor). TAK exosomes and dehydrocrenatine increased proliferation, while the miR-200c-3p mimic and astragaloside IV decreased proliferation. Different colors indicate the different treatment types.

**Table 1 ijms-26-02881-t001:** Demographic data and clinical features of TAK patients.

	Age (years)	Sex	Disease Duration (months)	Asthenia	Abdominal pain	Abortion	Hypertension	Stroke	ESR (mm/h)	hs-CRP (mg/L)	IL-6 (ng/L)	TNF-α (ng/L)	Numano Classification	ITAS-2010 Score
1	17	F	3	Y	N	N	Y	Y	7	5.44	5.9	9.6	II	7
2	46	F	34	Y	N	N	Y	N	80	8.35	2	9.3	V	8
3	15	M	1	Y	N	N	Y	Y	6	2.26	2	8.2	V	9
4	38	M	1	Y	N	N	Y	N	18	30.59	12.7	8.9	V	7
5	40	F	49	N	N	N	N	N	39	6.11	5.5	6.8	II	4
6	35	F	2	Y	N	N	Y	N	3	11.45	3.7	4	V	9
7	35	F	11	N	N	N	N	N	16	4.07	2.9	6.1	V	4
8	39	F	2	N	N	N	N	N	30	5.22	5.8	5	III	4
9	34	F	67	Y	N	N	Y	N	25	1.87	5.5	7.9	V	6
10	22	F	39	Y	N	N	Y	Y	23	36.79	8.5	5.7	V	11
11	42	F	114	N	N	N	N	N	20	2.42	59.5	5	V	1
12	33	F	44	Y	N	N	Y	N	25	21.56	8.6	7.1	V	10
13	37	F	14	Y	N	N	Y	N	75	10.01	31.56	3.7	V	9
14	22	F	9	Y	N	N	Y	N	23	21.92	19.53	16.8	V	8
15	20	F	3	N	N	N	Y	N	64	9.61	29.53	9.8	V	11
16	29	F	7	N	N	N	N	N	42	37.91	38.56	9.1	V	9
17	31	F	19	N	N	N	Y	N	51	9.35	11.76	11.5	V	6
18	45	F	59	N	N	N	N	N	15	18.44	11.34	13.4	V	7
19	32	M	22	N	N	N	Y	N	77	33.82	19.8	9.7	V	6
20	32	F	40	N	N	N	N	N	22	39.48	49.53	11.2	III	5
21	19	F	10	Y	N	N	N	N	39	42.02	7.5	8.5	V	6
22	37	F	57	Y	N	N	N	N	56	1.95	12.22	12.1	V	7
23	25	F	6	Y	N	N	N	N	75	4.75	14.55	6.4	V	9
24	29	M	13	N	N	N	N	N	84	3.1	9.5	6.5	II	6
25	23	F	17	N	N	N	Y	N	19	0.47	3.1	7.1	V	3
26	24	F	35	N	N	N	Y	N	59	0.66	3.9	9.2	V	1
27	22	F	48	N	N	N	Y	N	20	23.67	3.66	6	II	5
28	27	F	56	Y	N	N	N	N	27	9.13	8.77	10.1	V	4
29	23	F	13	Y	N	N	N	N	29	9.13	8.31	5.5	V	9
30	18	F	9	N	Y	N	Y	N	35	9.29	9.35	7.2	V	2

## Data Availability

The data used and/or analyzed during the current study are available from the corresponding author on reasonable request.

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
