# Peer review of "Downregulated Expression of miR-200c-3p in Plasma Exosome as a Potential Biomarker in Takayasu’s Arteritis"

_ijms, 2025, doi:10.3390/ijms26072881_

Round 1
Reviewer 1 Report
Comments and Suggestions for Authors
The study reports miR-200c-3p as being downregulated in plasma exosomes, serving as a biomarker for TAK. This validation study builds on previous findings where NGS was used to profile miRNAs, and miR-200c-3p is validated here in 12 TAK patients. Bioinformatics analysis suggests its potential role in cellular physiology, while in vitro experiments in human aortic endothelial cells highlight its involvement in the ERK1/2 pathway and effects on cell proliferation.
The methods used are logical and well-structured, and the discussion is consistent with the data and grounded in relevant literature.
Minor comments for refinement:
- Line 93 – The term "untreated TAK patients" is unclear and needs clarification.
- Figure 5C – It’s confusing whether the TAK-labeled lanes represent plasma exosomes from different TAK patients or plasma exosomes from one TAK patient treated with different miRNAs. Please clarify.
- miRNA treatments should also be tested on cells treated with healthy control (HC) exosomes to assess the baseline changes induced by the miRNAs.
- Figures 5D and E – How were the 100, 1000, and 10,000 EVs/cell calculated, and how was the treatment administered? This information should be added to the methods section.
Author Response
Comments 1: Line 93 – The term "untreated TAK patients" is unclear and needs clarification.
Response 1: Thank you for pointing this out. We agree with this comment. By "untreated TAK patients," we mean patients newly diagnosed with Takayasu’s arteritis (TAK) who have not yet received any medical (e.g., glucocorticoids, immunosuppressants) or surgical treatment for their condition. We have clarified this definition in the revised manuscript (Page 3, Line 94-96).
Comments 2: Figure 5C – It’s confusing whether the TAK-labeled lanes represent plasma exosomes from different TAK patients or plasma exosomes from one TAK patient treated with different miRNAs. Please clarify.
Response 2: Thank you for pointing this out. We agree with this comment. The TAK-labeled lanes in Figure 5C represent plasma exosomes isolated from different Takayasu arteritis (TAK) patients. We will clarify this in the figure legend in the revised manuscript (Page 8, Line 144-145).
Comments 3: miRNA treatments should also be tested on cells treated with healthy control (HC) exosomes to assess the baseline changes induced by the miRNAs.
Response 3: Thanks for your critical comments. We agree with this comment. We believe the control they are referring to is already included in Figure 5D, where cells were treated with healthy control (HC) plasma exosomes. This serves as the baseline comparison for the TAK plasma exosome treatments shown in Figure 5E. Could the reviewer please clarify if this addresses their concern, or if they are suggesting a different comparison?
Comments 4: Figures 5D and E – How were the 100, 1000, and 10,000 EVs/cell calculated, and how was the treatment administered? This information should be added to the methods section.
Response 4: Thanks for your important comments. We agree that the details of the exosome administration method should be included in the methods section for clarity. We will add a detailed description of the exosome treatment protocol to the 'Edu assay' subsection of the Methods section in the revised manuscript (Page 12, Lines 294-302).
Reviewer 2 Report
Comments and Suggestions for Authors
Du, Fang, et al. Report the downregulated expression of miR-200c-3p in plasma EVs as a potential biomarker in Takayasu’s arteritis. Although the research idea is interesting and has potential clinical implications, I found it very difficult to follow the manuscript, so I will recommend further editing to make the work clearer and more understandable. Language correction is really needed.
Besides that, my comments about the content itself are as follows.
Major comments:
- Abstract: It's really difficult to follow and I don't think it has the right format where the problem is presented and the conclusions are briefly described, it has methods information, and it's difficult to follow. It even looks like just copied/pasted from any AI tool.
- Introduction: Almost no background is provided in the state-of-art of miRNA and their relationship with this disease, only reference to a previous study of the authors.
- In TEM images some particles may be lipoprotein contamination. It would be important to add ApoA1 (or equivalent) to check for this contamination by WB.
- The qPCR validation of the previous sequencing results seems to not be working at all. 4 out of 5 DE miRNAs were not found as DE. Even the one that they found, miR-200c-3p, has a high pvalue, and a high overlap between both groups, with 3 patients falling into the healthy distribution.
- In the HAECs experiment, the authors claim that “These cells exhibited activated ERK1/2 pathway and JNK pathway, but not p38 MAPK pathway” with respect to TAK pEV treated cells, and that “This effect could be suppressed by miR-200c-3p mimic supplementation.” I can’t see this in the WB of Figure 5C, taking into account the GADPH band as a normalizer, all bands in the 4 TAK conditions are the same.
- Why there’s in methods a section for PBMCS and flow cytometry if not included in the results?
- Statistical methods are rudimentary and more strict cut-offs should be made.
Minor comments:
- Section 2.1 should be on methods, it’s not a result.
- The colours of figure 1c make impossible to see the points
- Figure 3 title is not a figure title
Comments on the Quality of English Language
The quality of English is bad, some sentences have no coherence with the ones before, typos, etc. It must be further edited.
Author Response
Comments 1: Abstract: It's really difficult to follow and I don't think it has the right format where the problem is presented and the conclusions are briefly described, it has methods information, and it's difficult to follow. It even looks like just copied/pasted from any AI tool.
Response 1: Thanks for your important comments. We understand the concern that the original abstract was difficult to follow and lacked a clear presentation of the problem and conclusions. We have revised the abstract (Page 1, Line 14-32). to address these points by:(1) Streamlining the background and focusing on the key research question;(2) Condensing the methods section to highlight only the core techniques;(3)Presenting the results more concisely and emphasizing statistical significance;(4)Revising the conclusion to clearly state the main findings and their potential implications. We believe the revised abstract now provides a more focused and reader-friendly summary of our study. We assure the reviewer that this abstract, and indeed the entire manuscript, was written by the authors and not generated using any AI tools.
Comments 2: Introduction: Almost no background is provided in the state-of-art of miRNA and their relationship with this disease, only reference to a previous study of the authors.
Response 2: Thanks for your critical comments. We appreciate the reviewer's comment regarding the need for more background information on miRNAs in the context of Takayasu's Arteritis (TAK). We acknowledge that the field is relatively nascent, and there is not a large body of published literature specifically addressing miRNA dysregulation in TAK pathogenesis compared to other diseases. Our original introduction focused primarily on our prior work due to the limited existing research directly investigating specific miRNAs and their mechanisms of action in TAK. In response to the reviewer's feedback, we have expanded the introduction to provide a more comprehensive overview of the role of miRNAs in vascular inflammation and remodeling, processes known to be central to TAK pathogenesis (Page 1, Line 42-45; Page 2, Line 46-63).
Specifically, we have:
- Included broader context on the involvement of miRNAs in other vasculitis and inflammatory vascular diseases. We have added information about the role of miRNAs in modulating inflammatory cytokine production in Behçet’s disease (Gu, F. et al.,2023) and the involvement of miR-21-3p in vascular smooth muscle cell proliferation in atherosclerosis (Jones et al., 2020)."
(2) Highlighted the emerging role of miRNAs as potential biomarkers and therapeutic targets in inflammatory diseases. We have cited a recent review by Jae, N. et al. (2020) highlighting the potential of circulating miRNAs as non-invasive biomarkers for disease activity and response to therapy in autoimmune disorders."
(3) Added a reference to our recent publication in Arthritis Research & Therapy. We have included our recent finding that exosome-derived miR-199a-5p plays a role in vascular remodeling and inflammatory infiltration in TAK, providing further evidence for the involvement of miRNAs in this disease.
We believe that these additions significantly strengthen the introduction and provide a more balanced and comprehensive overview of the current understanding of miRNAs in the context of vascular inflammation and TAK. We are confident that this revised introduction adequately addresses the reviewer's concerns.
Comments 3: In TEM images some particles may be lipoprotein contamination. It would be important to add ApoA1 (or equivalent) to check for this contamination by WB.
Response 3: Thank you for pointing this out. We thank the reviewer for raising the important point about potential lipoprotein contamination in our TEM images. We understand the concern that some observed particles might be lipoproteins rather than exosomes. In response to this concern, we performed a Western blot analysis to assess the presence of ApoA1, a major component of high-density lipoproteins (HDL), in our exosome preparations. The results of this Western blot are now included as Supplementary Figure S1 A. As shown in Supplementary Figure S1 A, a faint band corresponding to ApoA1 was detected in our exosome preparations. However, the intensity of this band was significantly lower than the ApoA1 signal detected in the material remaining after the exosome isolation process. This result suggests that while there may be a small degree of lipoprotein contamination in our exosome preparations, the majority of lipoproteins were effectively removed during the isolation process. The significantly lower ApoA1 signal in the exosome fraction compared to the remaining material indicates that lipoproteins are not a major component of the observed particles in our TEM images. Therefore, we are confident that the TEM images primarily represent exosomes, with minimal influence from lipoprotein contaminants (Page 2, Line 73-74).
Comments 4: The qPCR validation of the previous sequencing results seems to not be working at all. 4 out of 5 DE miRNAs were not found as DE. Even the one that they found, miR-200c-3p, has a high pvalue, and a high overlap between both groups, with 3 patients falling into the healthy distribution.
Response 4: Thanks for your important comments. We acknowledge that our initial qPCR validation with 12 patients showed limited concordance with the sequencing data, particularly with respect to the statistical significance and the degree of overlap between groups for miR-200c-3p. To address these concerns and improve the statistical power of our validation, We have expanded the sample size for our qPCR analysis, increasing the number of healthy controls (HC) to 20 and the number of treatment-naïve TAK patients from 12 to 30.
With the increased sample size, the qPCR validation now shows statistically significant differential expression for 1 out of the five DE miRNAs identified in our sequencing analysis. For miR-200c-3p (now presented in Figure 1C), the p-value has decreased to <0.001, and the overlap between the TAK patient and healthy control groups has been reduced, with fewer patients now falling within the healthy control range. We have included a revised Figure 1C and updated statistical analysis (Page 12, Line 312-321) in the manuscript to reflect these changes. The revised Figure 1C shows a clear separation between the TAK patients and healthy controls. Consistent with these findings, the ROC curve analysis presented in Figure S1B also demonstrates the strong ability of miR-200c-3p to discriminate between TAK patients and healthy controls.
Comments 5:In the HAECs experiment, the authors claim that “These cells exhibited activated ERK1/2 pathway and JNK pathway, but not p38 MAPK pathway” with respect to TAK pEV treated cells, and that “This effect could be suppressed by miR-200c-3p mimic supplementation.” I can’t see this in the WB of Figure 5C, taking into account the GADPH band as a normalizer, all bands in the 4 TAK conditions are the same.
Response 5: Thanks for your comments. We recognize that visual inspection of the blot alone may not fully capture the subtle changes in protein phosphorylation. To address this, we have performed quantitative densitometry analysis of the Western blot bands using ImageJ. The resulting data is presented in Supplementary Figure S2. As shown in Supplementary Figure S2, we observed the following: A statistically significant increase in the levels of phosphorylated ERK1/2 (pi-p44/42) and phosphorylated JNK (pi-SAPK/JNK) in HAECs treated with TAK pEVs compared to the control group (p < 0.05, Mann-Whitney U). Furthermore, the addition of the miR-200c-3p mimic significantly suppressed this increase in p-ERK1/2 and p-JNK levels (p < 0.05, Mann-Whitney U). As claimed, the levels of phosphorylated p38 MAPK (p-p38) were not significantly altered by any of the treatments.
Each data point in Supplementary Figure S2 represents the average of three independent experiments. We apologize that these changes were not readily apparent upon visual inspection of the original Western blot. We believe that the quantitative analysis provides more objective evidence to support our conclusions. We have clarified this by noting in the result that densitometry values are available in Supplementary Figure S2 (Page 8, Line 157-158).
Comments 6: Why there’s in methods a section for PBMCS and flow cytometry if not included in the results?
Response 6: Thank you for pointing this out. The PBMCs and flow cytometry methods were used to generate the data presented in Figure 3, which shows the circulating frequencies of Th1, Th2, Th17, and Th17.1 cells. In the legend of Figure 3, we will add that ‘Circulating frequencies of T helper cell subgroups of TAK patients and healthy controls were determined via flow cytometry’ (Page 5, Line 120-121).
Comments 7: Statistical methods are rudimentary and more strict cut-offs should be made.
Response 7: Thanks for your important suggestion. We acknowledge that the original statistical approach was relatively basic and that stricter cut-offs are warranted to ensure the robustness of our findings. In response to this feedback, we have taken the following steps to enhance the statistical rigor of our analysis:
- Increased Sample Size: As previously mentioned, we have increased the patient sample size from 12 to 30, the healthy control (HC) group was expanded to 20 individuals for the qPCR validation, which significantly improves the statistical power of the analysis.
- ROC Curve Analysis: We have performed Receiver Operating Characteristic (ROC) curve analysis to determine an optimal diagnostic cut-off for miR-200c-3p. The ROC curve (now presented in Figure S1B) demonstrates an Area Under the Curve (AUC) of 0.908 with a 95% Confidence Interval of 0.804-1.000. Based on this analysis, we have established a diagnostic cut-off of 2.0823 for miR-200c-3p, selected to maximize the Youden's Index, which exhibits a sensitivity of 1 and a specificity of 0.8 for distinguishing TAK patients from healthy controls (Page 2, Line 78-81).
Comments 8: Section 2.1 should be on methods, it’s not a result.
Response 8: Thanks for your very important comments. We agree that Section 2.1 presents methodological information and should be in the Methods section. We will move Section 2.1 to the Methods section in the revised manuscript (Page 10, Line 223-225).
Comments 9: The colours of figure 1c make impossible to see the points
Response 9: Thanks for this important point. We will revise the figure to use a color palette that provides better contrast and allows for clear visualization of all data points. This will be implemented in the revised manuscript.
Comments 10: Figure 3 title is not a figure title
Response 10: Thanks for your critical comments. We have revised the title to better reflect the data presented. The new title is: 'Figure 3. Spearman Correlation Analysis of miR-200c-3p Expression in Plasma Exosomes and Circulating Frequencies of T Helper Cell Subsets in Takayasu's Arteritis Patients.' (Page 5, Line 117-119).